# MACC1 Regulates LGR5 to Promote Cancer Stem Cell Properties in Colorectal Cancer

**DOI:** 10.3390/cancers16030604

**Published:** 2024-01-31

**Authors:** Müge Erdem, Kyung Hwan Lee, Markus Hardt, Joseph L. Regan, Dennis Kobelt, Wolfgang Walther, Margarita Mokrizkij, Christian Regenbrecht, Ulrike Stein

**Affiliations:** 1Experimental and Clinical Research Center, Charité—Universitätsmedizin Berlin and Max-Delbrück-Center for Molecular Medicine, Translational Oncology of Solid Tumors Research Group, 13125 Berlin, Germanydennis.kobelt@epo-berlin.com (D.K.);; 2Bayer AG, Research and Development, Pharmaceuticals, 13342 Berlin, Germany; 3JLR Life Sciences Ltd., A96 A8D5 Dublin, Ireland; 4German Cancer Consortium, 69120 Heidelberg, Germany; 5CELLphenomics GmbH, 13125 Berlin, Germany

**Keywords:** MACC1, LGR5, transcriptional regulation, stemness

## Abstract

**Simple Summary:**

We discovered a novel association of cancer and stemness. In particular, we demonstrate an MACC1—LGR5 link by transcriptional regulation of the crucial stemness gene LGR5 by MACC1, the inducer of tumor initiation, progression and metastasis. We show this regulation by using 2D and 3D cell culture models, in CRC-derived PDX mouse models and in human CRC patient samples. This study indicates that the metastasis inducer MACC1 acts not only as a cancer stem cell-associated marker, but also as a regulator of LGR5 expression and LGR5-mediated stem cell properties. Thus, interventional approaches targeting MACC1 would potentially improve further targeted therapies for CRC patients to eradicate CSCs and prevent cancer recurrence and distant metastasis formation.

**Abstract:**

Colorectal cancer (CRC) is one of the leading causes of cancer-related deaths worldwide. The high mortality is directly associated with metastatic disease, which is thought to be initiated by colon cancer stem cells, according to the cancer stem cell (CSC) model. Consequently, early identification of those patients who are at high risk for metastasis is crucial for improved treatment and patient outcomes. Metastasis-associated in colon cancer 1 (MACC1) is a novel prognostic biomarker for tumor progression and metastasis formation independent of tumor stage. We previously showed an involvement of MACC1 in cancer stemness in the mouse intestine of our MACC1 transgenic mouse models. However, the expression of MACC1 in human CSCs and possible implications remain elusive. Here, we explored the molecular mechanisms by which MACC1 regulates stemness and the CSC-associated invasive phenotype based on patient-derived tumor organoids (PDOs), patient-derived xenografts (PDXs) and human CRC cell lines. We showed that CD44-enriched CSCs from PDO models express significantly higher levels of MACC1 and LGR5 and display higher tumorigenicity in immunocompromised mice. Similarly, RNA sequencing performed on PDO and PDX models demonstrated significantly increased MACC1 expression in ALDH1(+) CSCs, highlighting its involvement in cancer stemness. We further showed the correlation of MACC1 with the CSC markers CD44, NANOG and LGR5 in PDO models as well as established cell lines. Additionally, MACC1 increased stem cell gene expression, clonogenicity and sphere formation. Strikingly, we showed that MACC1 binds as a transcription factor to the LGR5 gene promoter, uncovering the long-known CSC marker LGR5 as a novel essential signaling mediator employed by MACC1 to induce CSC-like properties in human CRC patients. Our in vitro findings were further substantiated by a significant positive correlation of MACC1 with LGR5 in CRC cell lines as well as CRC patient tumors. Taken together, this study indicates that the metastasis inducer MACC1 acts as a cancer stem cell-associated marker. Interventional approaches targeting MACC1 would potentially improve further targeted therapies for colorectal cancer patients to eradicate CSCs and prevent cancer recurrence and distant metastasis formation.

## 1. Introduction

Colorectal cancer (CRC) has a very high prevalence throughout the world. It is rated as the third most common cancer in males and the second in females. The high mortality of CRC is directly associated with metastatic disease. Therefore, it is necessary to discover molecular biomarkers for early diagnosis of tumors with high metastatic potential [1].

The metastasis-associated in colon cancer 1 (MACC1) gene was first identified by a genome-wide analysis of genes which are differentially expressed in human colon cancer tissues, metastases and normal tissues [2]. It is located on the human chromosome 7 (7p21.1) and encodes a protein of 852 amino acids. In contrast to normal colon mucosa, MACC1 displays highly elevated expression levels in primary and metastasized colon cancer tissues. It shows the highest expression in tumors or blood of those not yet distantly metastasized patients but who will metachronously metastasize or those patients who have already developed distant metastases. Additionally, the 5-year survival rate of patients with high MACC1 expression in their primary tumors is decreased from 80% to 15% [2,3].

MACC1 has also been shown to induce crucial metastasis-associated phenotypes, such as migration, invasion, cell dissemination, wound healing and proliferation in cell cultures. In xenografted, patient-derived and transgenic mouse models, MACC1 induces tumor initiation and progression, as well as liver and lung metastases [2,4]. We generated MACC1 transgenic mice and crossed them with adenoma-initiating APC^min^ animals. When we observed the transition to malignant carcinomas in vil-MACC1/APC^min^ mice, we found by transcriptomics of vil-MACC1/APC^min^ vs. APC^min^ mice a tremendous expression of stemness genes, such as Nanog (direct induction) and Oct4 (indirect induction), by MACC1 [4]. The clinical importance of MACC1 for metastasis prognostication, prediction and treatment planning has meanwhile been confirmed for more than 20 solid tumor types; certainly, this has been repeatedly shown for CRC [3].

Accumulating research in the last decades underlines the long-lasting hypothesis that human cancers can be considered as stem cell diseases, supporting that solid tumorigenesis, progression and chemoresistance/radiochemoresistance are initiated by a small population of cancer cells. According to the cancer stem cell (CSC) model, features of these small fractions of cancer cells are self-renewal and pluripotency. Tissues such as the intestinal epithelium continuously self-renew by the replication of this particular set of adult stem cells [5]. CSCs are also thought to be responsible for recurrence and metastasis [6]. Intestinal crypt base stem cells, from which CSCs derive, are located at the bottom of the crypt structures, and several molecules are used for their identification, including CD44, CD166, NANOG, Oct4, ALDH1 and LGR5 [6,7,8].

Leucine-rich repeat G-protein coupled receptor 5 (LGR5) was first identified by Hsu et al. in 1998 [9]. The human LGR5 gene is located at chromosome 12 at position 12q22-q23 and contains (like rat and mouse homologs) 907 amino acids and seven transmembrane domains. LGR5 protein is expressed in crypt-base populations which are able to develop into all differentiated lineages within the intestine [10]. LGR5 is a Wnt target gene and indicates dividing intestinal stem cells. Also, LGR5 is reported as a biomarker for the identification of stem cells in the small intestine and colon [11,12,13].

However, MACC1 expression in human colon CSCs and possible implications for stemness still remain elusive. Here, we report for the first time that the metastasis inducer MACC1 transcriptionally regulates LGR5 expression to promote cancer stem cell properties in CRC by using 2D and 3D cell culture models, CRC-PDX mouse models as well as human CRC patient samples.

## 2. Materials and Methods

### 2.1. Cell Culture and Organoids

The human colorectal cancer cell lines SW480 (ATCC #CCL-228), HCT116 (ATCC #CCL-247) and SW620 (ATCC #CCL-227) with low, moderate and high endogenous MACC1 expression were cultured in DMEM medium supplemented with 10% FCS. The cells were transduced for forced MACC1 expression or MACC1 knockdown, and MACC1 knockout cells were generated as described in [14].

The 3D organoids were maintained in basic medium by addition of 1 × GlutaMax (GIBCO #35050-038), 1 × Pen/Strep (PAA #P11-010), HEPES (SIGMA #H3537) in 10 mM, N-acetylcysteine (SIGMA #A9165) in 1 mM, 1 × N2 (GIBCO #17502-048) and 1 × B27 (GIBCO #17504-044) to Advanced DMEM/F-12 (GIBCO #12634-010). Complete medium was made by addition of hFGF (Cell Signaling #8910, 20 ng/mL), EGF (SIGMA #E9644, 50 ng/mL), amphotericin B (SIGMA #A2942, 2.5 µg/mL) and Y-27632 (SIGMA #Y0503, 10 µM) to basic medium.

### 2.2. PDX Generation and the Tumor Dissociation Procedure

PDX tumor models were generated as described by Schütte et al. [15], and the use of patient material for PDX was approved by the local Institutional Review Board of Charité—Universitätsmedizin (Charité Ethics; EA 1/069/11) and the ethics committee of the Medical University of Graz (Ethics Commission of the Medical University of Graz, Auenbruggerplatz 2, 8036 Graz, Austria) and was confirmed by the ethics committee of the St John of God Hospital Graz (23-015 ex 10/11).

For all in vivo experiments, the welfare of the animals was maintained in accordance with the general principles governing the use of animals in experiments of the European Communities and German legislation. The study was performed in accordance with the United Kingdom Coordinating Committee on Cancer Research (UKCCCR) regulations for the Welfare of Animals and with the German Animal Protection Law and was approved by the local responsible authorities, Berlin, Germany (issued to the EPO GmbH Berlin by the State Office of Health and Social Affairs, Berlin, Germany; approval no. G 0333/18).

For PDX dissociation, the PDX samples were freshly obtained from mice, washed in PBS and mechanically dissected into small pieces, which were added to a freshly prepared enzyme mix containing 2 mg/mL collagenase I, 2 mg/mL collagenase II, 640 µg/mL dispase and 500 Units DNase I in 5 mL of RPMI without FCS. An enzyme mix containing dissected tumor pieces was transferred to a gentleMACS C-tube and applied to the gentleMACS program m_impTumor_03.01. Then, the C-tubes were placed on the gentleMACS rotator in a 37 °C incubator for 30 min. After tumor digestion, the mixes were filtered twice through cell strainers, first with a 100 µm strainer and then with a 70 µm strainer. The filtrates containing single cells were centrifuged at 400× *g* for 5 min at room temperature to remove enzymes. A quantity of 1 mL of erythrocyte lysis buffer was added to the cell pellet for 10 min at room temperature in the dark to lyse the mouse erythrocytes. After incubation, 5 mL PBS was added and centrifuged at 550× *g* for 5 min at room temperature to remove the lysis buffer. The pellet was washed with PBS and the cell number was counted for the subsequent experiments.

Prior to transductions, organoids were enzymatically dissociated into single cells and re-seeded on ultra-low attachment plates (1 × 10^5^ cells/mL).

### 2.3. CD44-APC Tagging and Flow Cytometry Cell Sorting (FACS) Analysis

CD44-APC Mouse Anti-Human CD44 antibody (Clone G44-26, BD Pharmingen™, Heidelberg, Germany) and APC Mouse IgG2b κ Isotype Control (Clone 27-35, BD Pharmingen™) were added to a final dilution of 1:10 to PBS-washed cells. The concentration of cells was adjusted to a range from 1 × 10^6^ to 1 × 10^7^ cells per 100 µL PBS before adding the antibodies. The suspension was incubated for 45 min at 4 °C in the dark. After incubation, the cells were washed twice with PBS by repeating centrifugation at 300× *g* for 8 min and adding PBS. The cells were suspended in 500 µL PBS for flow cytometry cell sorting (FACS) analysis.

Cells tagged with the CD44-APC antibody were FACS sorted. The cell suspensions were filtered first to obtain single cells for FACS. Each FACS analysis was performed by gating for single cells, then for living cells via DAPI labeling. Then, the cells were sterile filtered through a 20 μm sieve for the collection of unsorted, CD44-low and CD44-high populations directly into PBS for the subsequent experiments.

### 2.4. Aldefluor Assay

Organoids and xenografts were processed to single cells and labeled using the Aldefluor assay according to the manufacturer’s (Stemcell Technologies, Cologne, Germany) instructions. The cells were sorted by FACS (BD FACS Aria II) for ALDH activity and DAPI to exclude dead cells.

### 2.5. RNA Sequencing

The cells were lysed in RLT buffer and processed for RNA using the RNeasy Mini Plus RNA extraction kit (Qiagen, Hilden, Germany). The samples were processed using Illumina’s TruSeq RNA protocol and sequenced on an Illumina HiSeq 2500 machine as 2 × 125 nt paired-end reads. The raw data in Fastq format were checked for sample quality using our internal NGS QC pipeline. Reads were mapped to the human reference genome (assembly hg19) using the STAR aligner (version 2.4.2a) [16]. Total read counts per gene were computed using the program “featureCounts” (version 1.4.6-p2) in the “subread” package, with the gene annotation taken from Gencode (version 19). The “DESeq2” Bioconductor package [17] was used for the differential expression analysis [18,19]. Array data are available in the ArrayExpress database (www.ebi.ac.uk/arrayexpress, assessed on 21 January 2024) under accession numbers ArrayExpress: E-MTAB-5209 and ArrayExpress: E-MTAB-8927.

### 2.6. In Vivo Tumorigenicity Assay

FACS-sorted PDO-derived cell populations of 1 × 10^3^ cells/animal were transplanted in matrigel subcutaneously (s.c.) into the left flank of 6–8-week-old anesthetized female NMRI nu/nu mice (n = 3 mice/sorted population). The mice were observed for a maximum of 105 days and maintained under sterile and controlled conditions (22 °C, 50% relative humidity). Tumor growth was measured in two dimensions with a caliper. Tumor volumes (TV in cm^3^) were determined by the formula: TV = (width^2^ × length) × 0.5. The work conducted with living animals (at EPO GmbH Berlin, Germany) was approved by local authorities (Landesamt für Gesundheit und Soziales, LaGeSo Berlin, Germany) under approval number H0023-09.

### 2.7. Quantitative Real-Time Polymerase Chain Reaction (qRT-PCR)

The cell culture RNA purification protocol from the GeneMATRIX Universal RNA Purification Kit (Roboklon, Berlin, Germany) was performed and the concentration of RNA was measured by using the NanoDrop™ 2000/2000c Spectrophotometer (Thermo Fisher, Darmstadt, Germany). For each sample, 50 ng of total RNA was reverse transcribed. Reverse transcription was performed with random hexamers in 5 mM MgCl_2_, 1 × RT buffer, 250 µM pooled dNTPs, 1 U/µL RNase inhibitor and 2.5 U/µL MuLV reverse transcriptase. The reaction was run at 42 °C for 15 min and 99 °C for 5 min, with subsequent cooling at 5 °C for 5 min.

The following primers for qRT-PCR were obtained from Biotez (Berlin, Germany): MACC1 (forward: 5′-TTCTTTTGATTCCTCCGGTGA-3′, reverse: 5′-ACTCTGATGGGCATGTGCTG-3′), LGR5 (forward: 5′-GAGTTACGTCTTGCGGGAAAC-3′, reverse: 5′-TGGGTACGTGTCTTAGCTGATTA-3′), NANOG (forward: 5′-TTTGTGGGCCTGAAGAAAACT-3′, reverse: 5′-AGGGCTGTCCTGAATAAGCAG-3′), S100A4 and G6PD [14]. Each qRT-PCR reaction was performed in triplicate and in parallel to the cDNA quantification of the housekeeping gene. Each PCR reaction was performed in a total volume of 10 µL, containing 8 µL of reverse transcript and 2 µL of primer/GoTaq^®^ qPCR Master Mix, in 96-well-plates in the LightCycler^®^ 480. The PCR protocol for Promega SYBR Green-based qRT-PCR comprised a pre-incubation step at 95 °C for 2 min followed by 45 cycles of denaturation at 95 °C for 7 s, annealing at 60 °C for 10 s and elongation at 72 °C for 5 s. Data analysis was performed with LightCycler 480 Software release 1.5.0 SP3 (Roche, Penzberg, Germany). For each qRT-PCR reaction, a mean of the triplicate was calculated. Each mean value of the expressed gene was normalized to the respective mean amount of the housekeeping cDNA.

### 2.8. Tumor Sphere Formation Assay

Cells were counted and diluted in the corresponding medium to a concentration of 0.5 cells/µL (1 × 10^2^ cells in 200 µL). Edge wells in ultra-low-attachment 6-well plates were filled with PBS to minimize medium loss via evaporation. A quantity of 200 µL of cell suspension was added to each well and the plates were sealed with Parafilm to avoid evaporation of the medium. The plates were incubated at 37 °C for 7 days. The number of spheres formed was counted using 40× magnification. The results were represented as a percentage of the number of tumor cell spheres present divided by the initial number of cells seeded. While counting, only the solid circular spheres were counted, excluding aggregated cells.

### 2.9. Clonogenic Formation Assay

A total of 5 × 10^2^ cells were plated per well in 6-well plates in the corresponding medium and placed in the incubator at 37 °C for 7 days (except SW480 cells, as the colony formation required 14 days for proper colony formation and analysis). After the respective incubation times, the media were aspirated, and 800 µL of crystal violet/formaldehyde mix was added to each well and incubated for 40 min at room temperature. After removal of the crystal violet/formaldehyde mix, the plates were washed in water 3 times by submerging the plates for complete removal of residual dye. The plates were placed upside-down overnight for complete evaporation of water and for prevention of any watermarks on the wells, which may have interfered with the subsequent steps. Pictures were taken with the FluorChem Q system (Biotechne, Minneapolis, MN, USA) and analyzed using the Colony Assay program in Image J (NIH, Bethesda, MD, USA). The threshold of exposure was always adjusted before the complete analysis to ensure elimination of background signals.

### 2.10. Chromatin Immunoprecipitation (ChIP)

ChIP was performed using the Magna ChIP™ HiSens Kit (Merck, Darmstadt, Germany) according to the manufacturer’s instructions. Cell lysates were sonicated for 24 pulses at 100% output, and immunoprecipitation of the DNA-protein complexes was performed using Magna ChIP A/G beads (Merck, Darmstadt, Germany) overnight at 4 °C followed by DNA isolation. Mouse IgG and anti-RNA polymerase II antibodies (both obtained from Merck Millipore, Burlington, MA, USA) were used as negative and positive controls, respectively. The binding of MACC1 to the LGR5 gene promoter (LGR5 gene promoter forward primer: 5′-TCACTTCGACTTCCTCACCC-3′ and reverse primer: 5′-CACTGTCTGGCTCGCTTTTG-3′) was evaluated via specific qRT-PCR.

### 2.11. Patient Sample Analysis

We analyzed tissue specimens from 59 patients suffering from CRC (with informed written consent and approval by Charité Ethics Committee, Charité University Medicine, Berlin, Germany), which were used in our previous study [4]. All patients did not have a history of familial colon cancer and did not suffer from a second tumor of the same or a different entity. All patients were staged I, II or III (not distantly metastasized at the time point of surgery). They were previously untreated, and the patients’ tumors were surgically R0-resected (complete resection with no microscopic residual tumor). Ethical approval and patient consent to participate: All analyses were carried out in accordance with the guidelines approved by the institutional review board, number AA3/03/45, of the Charité—Universitätsmedizin Berlin, Germany. All patients gave written informed consent and the authors complied with all relevant ethical regulations for research involving human participants.

### 2.12. Statistical Analysis

All calculations and statistical analyses were performed with GraphPad Prism version 5.01. Comparisons of two groups were performed by two-tailed, paired Student’s *t*-tests. More than two groups were compared using ANOVA and appropriate post-tests. Correlations between MACC1 and LGR5 in cell lines and patient samples were evaluated by using the Spearman-rho test. All tests were two-sided, and *p* values ≤ 0.05 were considered to be statistically significant (* *p* < 0.05, ** *p* < 0.01, *** *p* < 0.001, **** *p* < 0.0001). Error bars represent the standard deviations. Whisker boxes show the means, minimums, maximums, and 1st and 3rd quartiles.

## 3. Results

### 3.1. Expression of MACC1 and LGR5 Are Elevated in CD44-High Stem Cell-Enriched Populations of Patient-Derived 3D Organoids

We started our studies with the generation of CRC patient-derived organoids (PDOs), as described by Schütte et al. 2017 [15]. To characterize the stem cell-enriched cell population, we sorted for CD44-low, -medium and -high cell populations of eight PDOs. Within these cell populations, we measured RNA levels of MACC1, LGR5, Muc2 (Mucin2, intestinal, [20]) and CK20 (intestinal, [21]) by qRT-PCR. Interestingly, the expression levels of MACC1 (*p* = 0.01) and LGR5 (*p* = 0.0002) were elevated in most CD44-high cell populations (Figure 1).

We then knocked down MACC1 in these PDO cells and observed a subsequent LGR5 downregulation, whereas CD44 was not affected. Further, we investigated the effect of MACC1 knockdown on phenotypical stemness features and observed a clearly reduced tumor sphere formation (Figure 2).

### 3.2. Tumorigenicity of CD44-High/CD166+ Subpopulations of PDO-Derived PDXs

Subpopulations selected from PDOs were chosen for subcutaneous transplantation of CD44-low/CD166+ or CD44-high/CD166+ subpopulations into three mice (CD166 was used to identify a more specific stem cell population). We additionally measured CD166 (stem cell marker) and S100A4 (MACC1-regulated metastasis executer; [14]). Tumorigenesis and increased tumor volumes were observed in those mice transplanted with CD44-high/CD166+ cells also showing high MACC1 and LGR5 levels (vs. CD44-low/CD166+ cells). Interestingly, S100A4—a transcriptional MACC1 target [14]—was also found to be elevated when MACC1 expression was increased (Figure 3).

### 3.3. Expressions of MACC1 and LGR5 Are Elevated in ALDH1-Positive Stem Cell-Enriched Populations of Patient-Derived 3D Organoids

In order to prove the link between MACC1 and stemness properties, five PDOs were sorted for ALDH1 (aldehyde dehydrogenase 1; stem cell marker and therapeutic target; [22,23]) -negative and -positive subpopulations. Interestingly, MACC1 as well as LGR5 mRNA levels were increased in ALDH1-positive cells (vs. ALDH1-negative cells; Figure 4a). Further, we also generated patient-derived xenografted mice (PDXs) from these PDOs. When sorting for ALDH1-negative and -positive subpopulations, we observed an increased expression of MACC1 in four out of five models in ALDH1-positive cells (Figure 4b). Taken together, RNA sequencing performed on PDO and PDX models demonstrated increased MACC1 expression in ALDH1-positive CSCs, highlighting its involvement in cancer stemness.

### 3.4. The MACC1—Stemness Marker Link in CRC 2D Cell Lines

Next, we tested the link between MACC1 and stemness in the 2D CRC cell lines SW480 (Figure 5a), HCT116 (Figure 5c,d) and SW620 (Figure 5b) with low, moderate and high endogenous MACC1 expression. Forced expression of MACC1 in endogenously low-MACC1-expressing SW480 cells led to increased expression of NANOG and LGR5. Knockout of MACC1 in the endogenously high MACC1 SW620 cells reduced NANOG and LGR5. Additionally, the expression reduction in LGR5 upon MACC1 knockout was rescued by overexpressing MACC1 in these knockout clones. This validated the previous findings, where we showed MACC1-dependent expression regulation of stem cell factors. In the endogenously moderately expressing HCT116 cells, forced expression of MACC1 resulted in increased NANOG and LGR5 expression, whereas MACC1 knockout diminished NANOG and LRG5 levels. Thus, we confirmed the MACC1—stemness link also in 2D CRC cell lines (Figure 5a–d).

### 3.5. The MACC1—Stemness Link in CRC 2D Cell Lines: Phenotypic Assays

Next, we tested if stemness phenotypes in CRC 2D cell lines are influenced by MACC1 expression manipulation. We analyzed clonogenicity in SW480/EV vs. SW480/MACC1, HCT116/EV vs. HCT116/MACC1 and HCT116/Cas9 vs. HCT116/MACC1-KO cells. Colony numbers were significantly increased when MACC1 expression was increased (Figure 6a). Then, we analyzed tumor sphere formation of HCT116/GFP vs. HCT116/MACC1 and SW620/Cas9 vs. SW620/MACC1-KO cells. We observed significantly increased tumor sphere formation in high-MACC1-expressing cells (Figure 6b). Knockout of MACC1 led to a decrease in stem cell properties.

### 3.6. Back Translation of the MACC1—LGR5 Association in CRC-PDX Models

Four tumors from CRC-PDXs were dissociated and CD44-low and CD44-high cell populations were sorted. Thereafter, we determined the specific gene expression levels for MACC1 and LGR5 in unsorted, CD44-low and CD44-high PDX-derived cells. Interestingly, we observed significantly higher MACC1 and LGR5 expression levels in CD44-high cells, confirming the discovered MACC1-LGR5 association also in passaged CRC-PDX-derived cells (Figure 7).

### 3.7. MACC1 Increases Cancer Stemness via Transcriptional Activation of the LGR5 Target Gene

We then explored if MACC1 directly regulates the expression of LGR5 via the LGR5 promoter (Figure 8a). Using ChIP, we found a clear binding of MACC1 to the LGR5 promoter in SW620 cells, making LGR5 a direct target gene of MACC1 (Figure 8b). This finding substantiates the previous data which showed MACC1-dependent regulation of stemness and CSC-like phenotypes in both PDO and PDX models, as well as in 2D CRC cell-line models.

### 3.8. Correlation of MACC1 with LGR5 Expression in Different CRC Patient Cohorts

Finally, we proved the correlation of MACC1 with LGR5 expression in different cohorts of CRC patients (Figure 9). We started with a first cohort of 59 CRC patients, for which MACC1 expression is already published [2] and for which we now measured the LGR5 expression in the 59 patients. We found a significant MACC1-LGR5 correlation (*p* < 0.0001) with an r = 0.5169 (Figure 9a). In additional already published CRC cohorts [24,25,26], we also found significant MACC1-LGR5 expression correlations (*p* = 0.0259, *p* = 0.0473 and *p* < 0.0001) with Spearman r = 0.3031, r = 0.5659 and r = 0.7273 values (Figure 9b). Strikingly, MACC1 and LGR5 expression showed moderate to strong correlation in CRC patients from different cohorts. Additionally, patients with high MACC1 mRNA expression had a significantly higher expression of LGR5 mRNA, as determined by qRT-PCR.

## 4. Discussion

Here, we report for the first time the link of MACC1, which is known to induce tumor initiation, progression and metastasis, with stemness by transcriptional expression activation of LGR5, a central stemness gene. We proved this link by using 2D and 3D cultures of human CRC cell lines, CRC-PDX mouse models and human CRC patient samples. First, we found higher expressions of MACC1 and stemness genes like LGR5 in CD44-high and ALDH1-positive stem cell populations. Interestingly, we verified an association of MACC1 and LGR5 by direct binding of MACC1 as a transcription factor to the LGR5 gene promoter. We supported this link by forced expression, knockdown or knockout of MACC1, followed by subsequent corresponding expression of stemness genes such as LGR5, as well as phenotypic features like the initiation of tumor sphere formation and clonogenicity. This newly discovered context was confirmed by MACC1-LGR5 expressions in four different CRC patient cohorts, newly described or publicly available. Taken together, MACC1 promotes cancer stem cell-like properties in CRC via employment of LGR5 as a novel signaling mediator.

In general, MACC1 has already been described in the stemness context in different cancer entities. In CRC, this association was analyzed in several studies [27,28,29,30], demonstrating the involvement of FoxA3, miR-3163 and DCLK1 able to phosphorylate MACC1. Targeting MACC1 in CRC not only affects known MACC1-induced features such as migration and invasion, but also influences stemness-associated phenotypes. In cervical cancer, MACC1 regulates the AKT/STAT3 signaling pathway to induce migration and invasion, but also cancer stemness [31]. In lung cancer, long noncoding RNA MACC1-AS1 was described to promote stemness through promoting UPF1-mediated destabilization of LATS1/2 [32]. The same long noncoding RNA MACC1-AS1 was also described to promote stemness by antagonizing miR-145 in hepatocellular carcinoma cells [33], via suppressing miR-145-mediated inhibition of the SMAD2/MACC1-AS1 axis in nasopharyngeal carcinoma [34], and promotes stemness and chemoresistance through fatty acid oxidation in gastric cancer [35].

Merlos-Suarez et al. reported the intestinal stem cell signature to identify CSC by LGR5 and to predict disease relapse. Interestingly, they noted the tumor-promoting and metastasis-inducing MACC1 to be upregulated in LGR5-positive stem cells of the mouse intestine (supplement in [13]). In this study, we enriched stem cell populations by CD44 and also by ALDH1. We not only showed the association of MACC1 and LGR5 expression, but also demonstrated the molecular mechanism by binding of MACC1 to the gene promoter of LGR5 initiating its expression and inducing stem cell properties like tumor sphere formation.

Since MACC1 is decisive for metastasis induction, the importance of the stemness gene LGR5 in cancer metastasis is crucial to extract. The causal link of LGR5 and cancer metastasis has been demonstrated by de Sousa e Melo et al. [36]. The authors developed an orthotopic mouse model and injected organoids directly into the colon mucosa, leading to tumors disseminated primarily to the liver. FACS analysis revealed enrichment of LGR5-GFP+ cells in micro-metastasis (5 weeks after injection) compared to both primary tumors and macro-metastasis (6 weeks after injection), suggesting that dissemination and/or colonization of distant sites may originate from LGR5+ CSCs. Further, the requirement of LGR5+ CSCs was not only shown for the development of liver metastasis, but also for the maintenance of established liver metastasis.

Indeed, there are several biological features supporting the MACC1-LGR5 link, i.e., the involvement of MACC1 and of LGR5 in clathrin-mediated endocytosis [37,38], their localization at the tumor invasion front [39,40], their role in core clock regulation [41,42] and their impact on craniofacial development [43,44]. Further, the regulation of both genes shares defined signaling pathways, such as the Wnt signaling pathway [45,46]. Importantly, both genes are of prognostic value for CRC [3,47], and, in addition, several other solid tumor entities are reported with crucial roles of both genes for patient prognosis, such as gastric, breast, ovarian, pancreas, intrahepatic cholangiocarcinoma, neuroblastoma and nasopharyngeal cancer.

## 5. Conclusions

Taken together, this newly discovered MACC1—LGR5 association contributes to better understanding of the stemness-tumor progression/metastasis link. This study indicates that the metastasis inducer MACC1 acts not only as a cancer stem cell-associated marker, but also as a regulator of LGR5 expression and LGR5-mediated stem cell properties. Since CSCs are believed to be responsible for tumor metastasis and relapse, interventional combinatorial approaches targeting MACC1 [48,49] and LGR5 [50,51] will potentially improve further targeted therapies for CRC patients to eradicate CSCs and prevent cancer recurrence and distant metastasis formation.

## Figures and Tables

**Figure 1 cancers-16-00604-f001:**
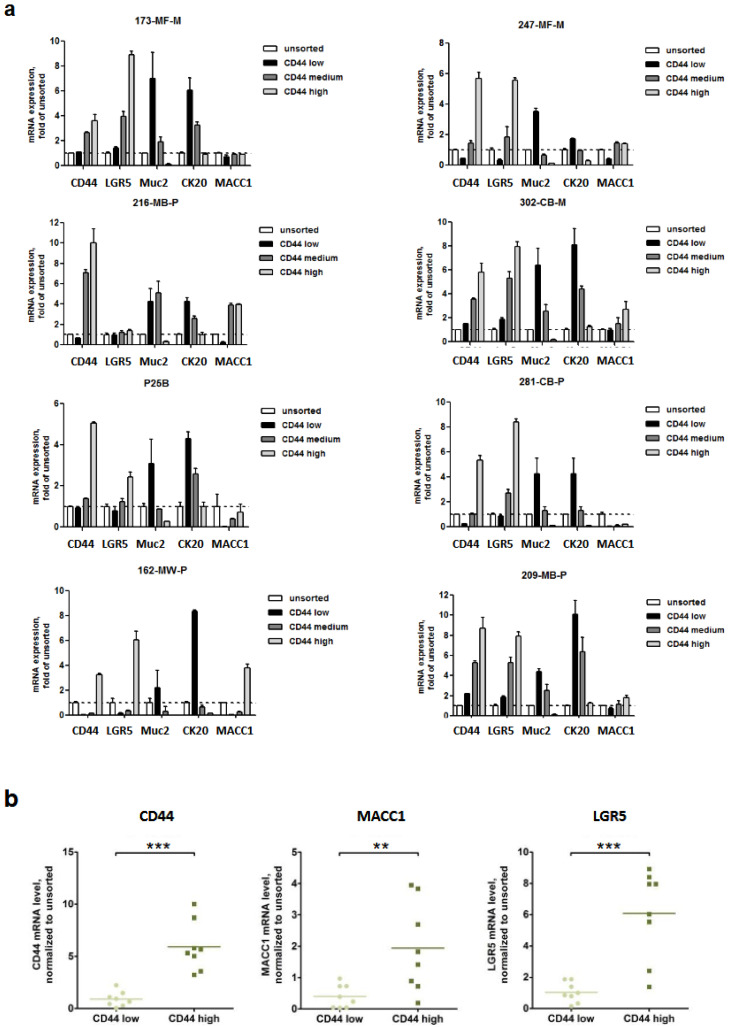
CD44-enriched CSCs from the PDOs expressed higher levels of MACC1 and LGR5. (**a**) mRNA expression analyses of a total of eight PDO samples, performed directly after FACS. (**b**) Summarized expression of single qRT-PCR values. (** = *p* < 0.01, *** = *p* < 0.001).

**Figure 2 cancers-16-00604-f002:**
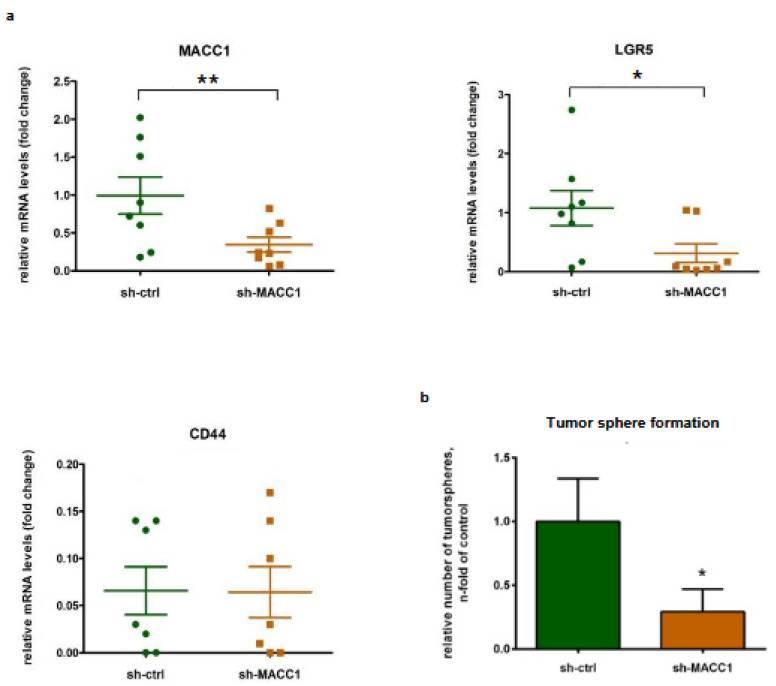
Knockdown of MACC1 in PDOs resulted in reduced LGR5 levels and cancer stemness phenotype. (**a**) Shown here is the effect of MACC1 knockdown of a total of four patient samples. (**b**) Illustrated here is the impact of MACC1 knockdown on the tumor sphere formation ability of a total of four PDO models. (* = *p* < 0.05, ** = *p* < 0.01).

**Figure 3 cancers-16-00604-f003:**
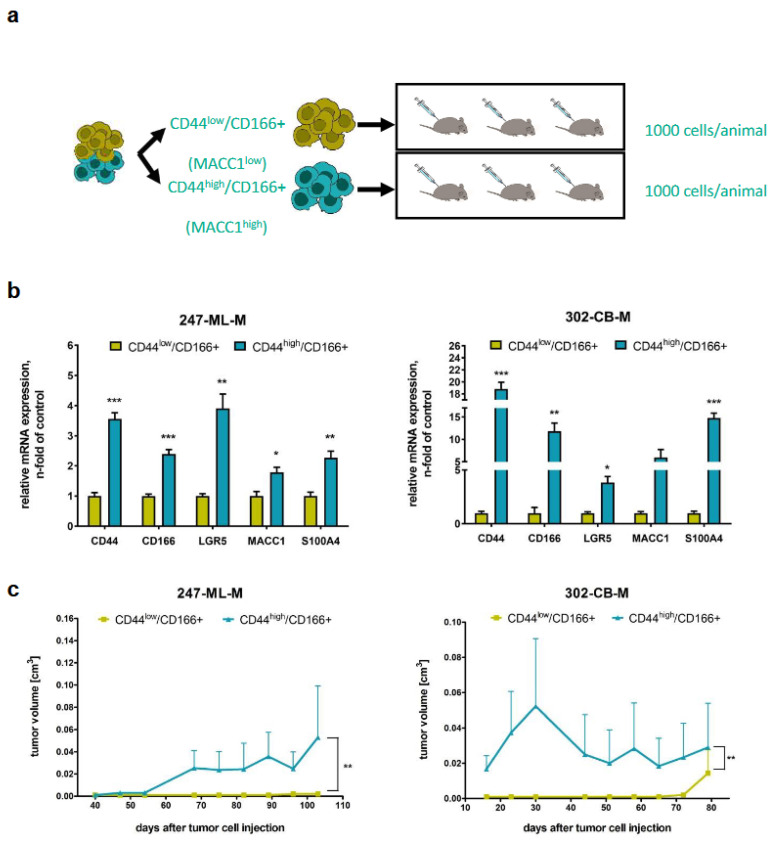
CD44-high/CD166+ stem cell-enriched populations display higher tumorigenicity in vivo, as illustrated by stem cell populations of two PDOs. (**a**) Experimental design. (**b**) qRT-PCR of CD44, CD166, LGR5, MACC1 and S100A4 in CD44-low/CD166+ and CD44-high/CD166+ stem cell-enriched populations. (**c**) Tumor volumes as measured in the respective mouse models. (* = *p* < 0.05, ** = *p* < 0.01, *** = *p* < 0.001).

**Figure 4 cancers-16-00604-f004:**
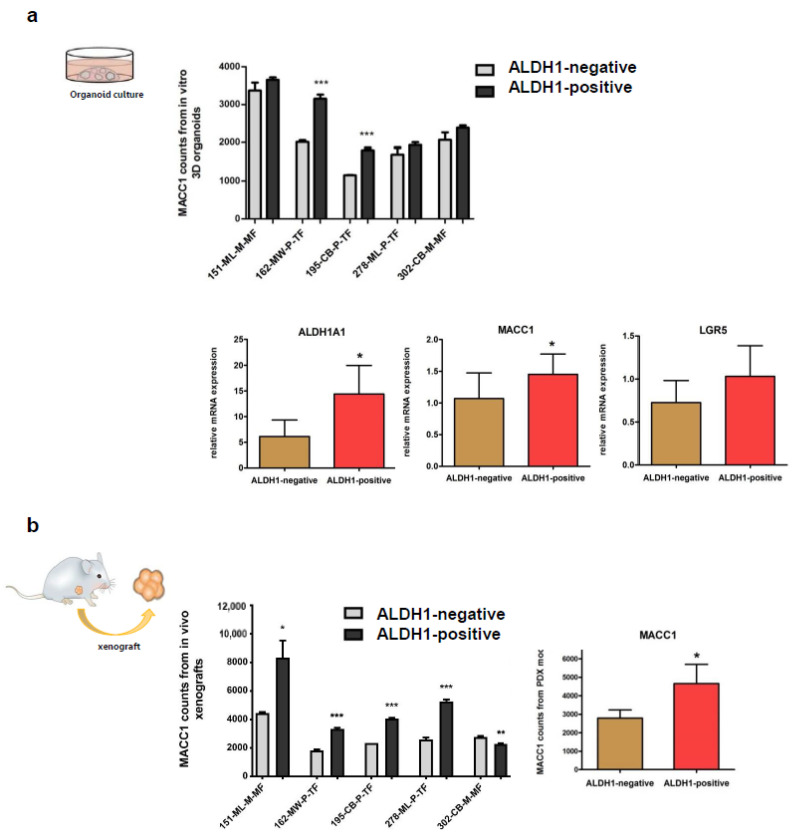
ALDH1-positive-enriched CSCs express increased MACC1 and LGR5 in PDO as well as PDX models. (**a**) Expression of ALDH1A1, MACC1 and LGR5 in ALDH1-positive and ALDH1A1-negative subpopulations of five PDOs. (**b**) RNA-sequencing generated FPKM values for MACC1 in ALDH1A1-positive and ALDH1A1-negative subpopulations of these five PDX models (n = 4 separate cell preparations). (* = *p* < 0.05, ** = *p* < 0.01, *** = *p* < 0.001).

**Figure 5 cancers-16-00604-f005:**
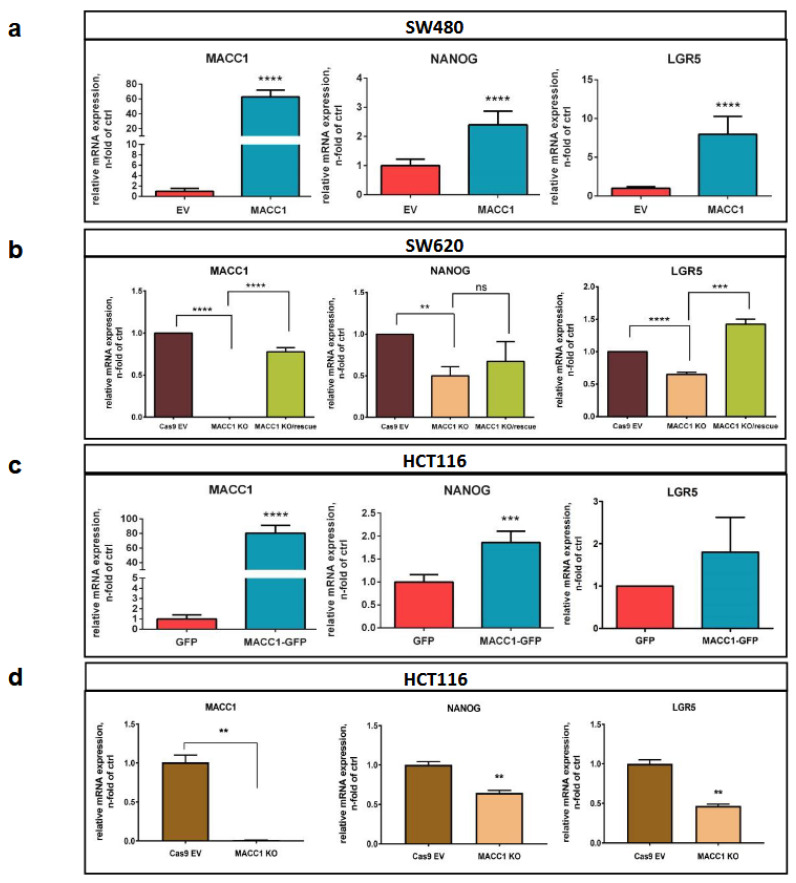
MACC1 regulates expression of cancer stemness genes in CRC cell lines. Expression levels of NANOG and LGR5 correspond to MACC1 levels in MACC1-manipulated (forced ectopic vs. knockout expression) SW480 (**a**), SW620 (**b**) and HCT116 (**c**,**d**) CRC cells. Rescuing MACC1-KO cells with forced expression of MACC1 almost normalized the expression levels of LGR5 in SW620 cells. (** = *p* < 0.01, *** = *p* < 0.001, **** = *p* <0.0001).

**Figure 6 cancers-16-00604-f006:**
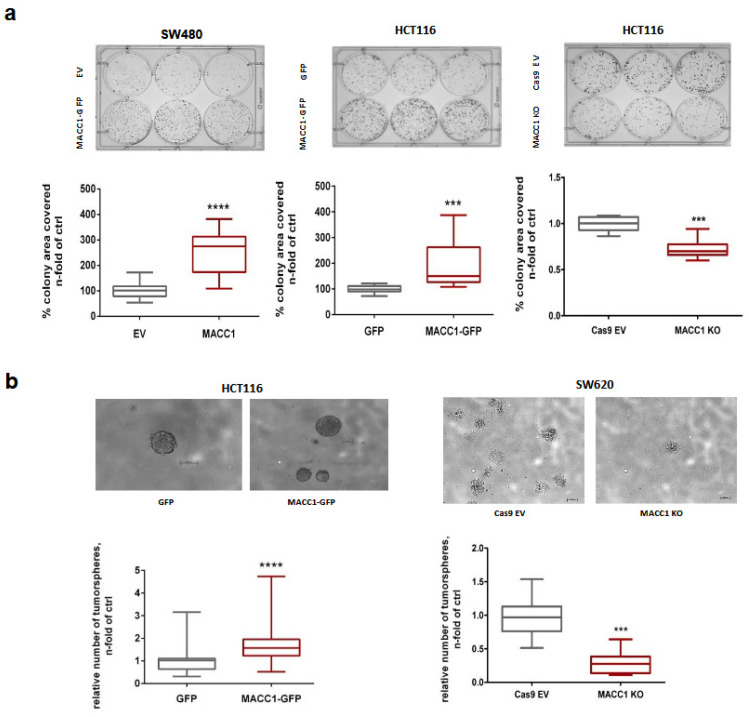
Clonogenicity and tumor sphere formation in MACC1-manipulated 2D CRC cell lines. (**a**) Clonogenicity in SW480/EV vs. SW480/MACC1, HCT116/EV vs. HCT116/MACC1 and HCT116/Cas9 vs. HCT116/MACC1-KO cells. Colony numbers were higher when MACC1 expression was increased. (**b**) Tumor sphere formation in HCT116/GFP vs. HCT116/MACC1 and in SW620/Cas9 vs. SW620/MACC1-KO cells. We observed increased tumor sphere formation in high-MACC1-expressing cells. (*** = *p* < 0.001, **** = *p* <0.0001).

**Figure 7 cancers-16-00604-f007:**
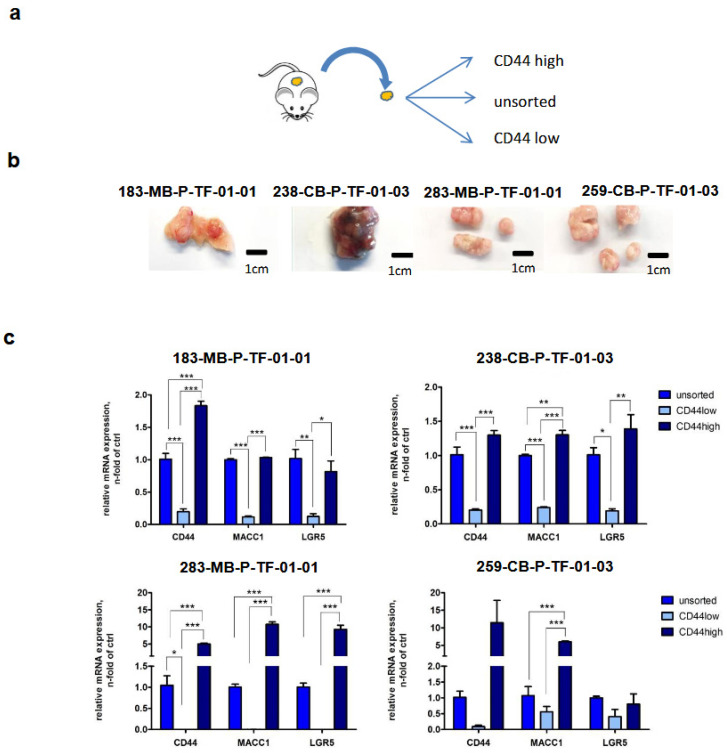
Expression analyses of MACC1 and LGR5 in CD44-low and CD44-high cell populations derived from CRC-PDX. (**a**) Experimental design. (**b**) Analyzed PDOs. (**c**) Expression of CD44, MACC1 and LGR5 in CD44-sorted cell population of PDOs. (* = *p* < 0.05, ** = *p* < 0.01, *** = *p* < 0.001).

**Figure 8 cancers-16-00604-f008:**
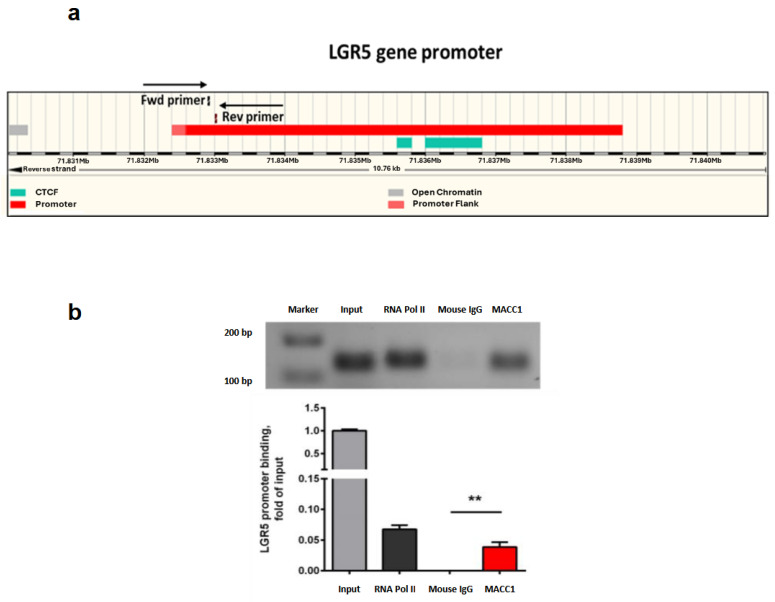
MACC1 increases cancer stemness via transcriptional activation of the LGR5 target gene. (**a**) Schematic representation of the full-length human LGR5 gene promoter region and the binding region for MACC1 detected by the respective primers. (**b**) ChIP for analysis of direct binding of MACC1 to the human LGR5 promoter region in SW620 cells. (** = *p* < 0.01).

**Figure 9 cancers-16-00604-f009:**
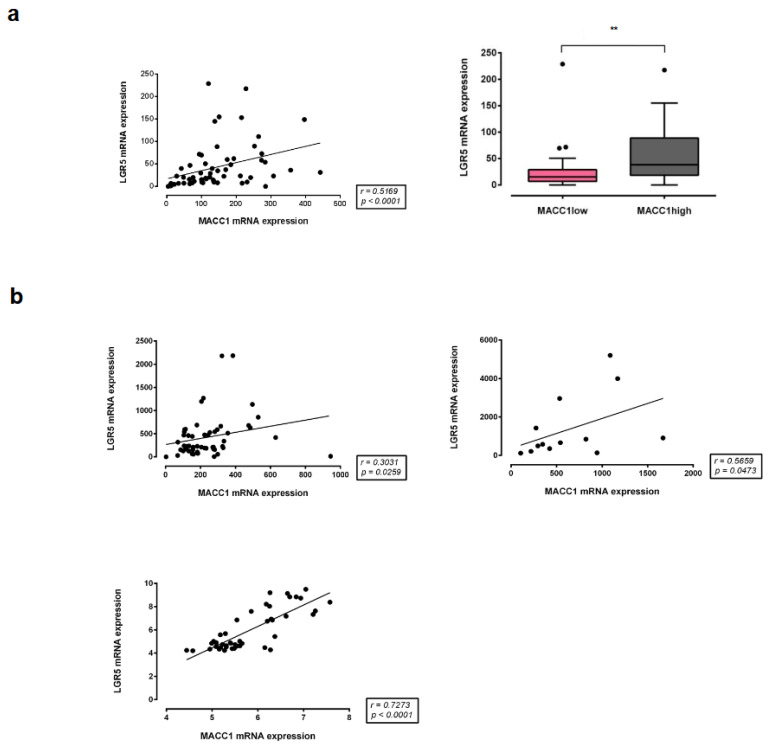
MACC1 expression correlates with LGR5 in CRC patients. Correlation analyses of MACC1 and LGR5 from CRC patient tumors. (**a**) MACC1 and LGR5 mRNA levels were determined by qRT-PCR and calculated as folds of the calibrator. Shown here is the LGR5 expression analysis of MACC1-low and MACC1-high patients. Classification of patients as low and high MACC1 expressers was performed using MACC1 median expression as a cutoff. (**b**) Datasets from other research groups were obtained from microarray analyses followed by normalization and filtering of the raw data using different algorithms. Correlation analysis was performed with nonparametric Spearman correlation [24,25,26]. ** *p* < 0.01.

## Data Availability

Data are available on request from the corresponding author.

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
