# Peer review of "MACC1 Regulates LGR5 to Promote Cancer Stem Cell Properties in Colorectal Cancer"

_cancers, 2024, doi:10.3390/cancers16030604_

Round 1

Reviewer 1 Report

Comments and Suggestions for Authors

The manuscript explores the association between MACC1 and LGR5 in colorectal cancer (CRC). The study employs a comprehensive approach, involving human CRC cell lines, patient-derived 3D organoids, CRC patient samples, and mouse models (PDX). They unveiled a novel association between MACC1 and LGR5, shedding light on the interplay between metastasis, stemness, and tumor progression in CRC. The findings suggest potential therapeutic strategies targeting both MACC1 and LGR5 to address the challenges posed by cancer stem cells in CRC patients. To provide a more comprehensive understanding, the following analysis could be incorporated:

1. While the study establishes MACC1 as a transcriptional activator of LGR5, it may benefit from a more in-depth exploration of downstream signaling pathways and molecular mechanisms involved in MACC1-mediated stemness. Additionally, it could provide a more comprehensive understanding to explore downstream targets and signaling pathways influenced by MACC1-LGR5 interaction.

2. The authors could assess the sensitivity of MACC1-high and MACC1-low cells to existing therapies or experimental agents targeting stemness pathways.

3. If the data is available, what is the correlation between MACC1 and LGR5 expression levels and patient survival outcomes.

Minors:

1. The Figure formats need to be consistent. The authors should improve the quality of presentation.

2. Some figures are missing statistical analysis, including Fig 1a, Fig 3b, 3c, Fig 5d. Also, the authors mixed the numeric values and * for the figures, the format needs to be consistent.

3. “we investigated the effect of MACC1 knockdown on phenotypical stemness features and observed a clearly reduced tumor sphere formation (Figure 2).” However, Figure 2 actually represented reduced LGR5 with MACC1 KD.

4. Any representative images for Fig 3c?

Author Response

see uploaded file

Reviewer 2 Report

Comments and Suggestions for Authors

 This is an interesting study that links MACC1 to colon cancer cell stemness via LGR5. Relevant primary colon cancer cultures and established cell lines as well as data generated from patient samples are used and function analyses using knockdown or knockout of MACC1 or overexpression are used to generate mechanistic support.

The models used, the data generated and their interpretation are presented very clearly and referenced to existing literature appropriately. Before publication, the author should correct some of the text and figure legends as outlined below.

In summary, this is an interesting study with solid data. Some small edits are needed.

Edits to text:

page 6, 3rd para: “… and observed a clearly reduced tumor sphere formation (Figure 2). 

Figure 2 does not show the tumor sphere formation data. Please correct.

page 6, para 3.3 and Figure 4:

…RNA sequencing performed on PDO and PDX models demonstrated increased MACC1

The data in Figure 4 also appear to be from RNAseq analysis. State that in the legend. Provide the units for the counts that are shown on the y-axes (e.g. transcripts per Mio reads (TPM)).

p. 7 Discussion, line 4: “using human 2D and 3D CRC cell lines”

should read “using 2D and 3D cultures of human CRC cell lines”

minor points:

Authorship attribution:

C. Regenbrecht #5 should be corrected to

C. Regenbrecht #4, i.e. Cellphenomics;  there is no #5

p. 1, Simple Summary: “in 2D and 3D CRC cell lines,” should be “in 2D and 3D cultures of CRC cell lines”

p. 3, line 1: “by using 2D- and 3D-cell models,” “by using 2D- and 3D-cell culture models”,

Edits to Figure Legends:

Figure 1. “CD44-enriched CSCs from the PDOs expressed higher levels of MACC1 and LGR5 and increased tumor sphere formation

Figure 1 shows mRNA expression; it does not show tumor sphere formation.

Figure 1 should read: “CD44-enriched CSCs from the PDOs expressed higher levels of MACC1 and LGR5

Figure 2. “Knock down of MACC1 in PDOs resulted in reduced LGR5 level. Shown here is the tumor sphere formation ability of a total of 4 patient samples upon MACC1 knockdown.

Figure 2 shows mRNA expression of MACC1, LGR5 and CD44 upon sh-MACC1 knockdown. The figure does not show tumor sphere formation. Should rephrase the legend accordingly.

Figure 4. “… as well as PDX models. (a) Expression of ALDH1,” should read “… as well as PDX models. (a) Expression of ALDH1A1

The expression is for ALDH1A1 mRNA, as indicated in the title of the panel (a).

The data  are from RNAseq analysis; should be stated in the legend. Also, provide the units for the counts that are shown on the y-axes (e.g. transcripts per Mio reads (TPM) or other conversions used).

Figure 5. “… and CSC-associated phenotype in CRC cell lines.

should be corrected. The figure shows mRNA expression and not phenotype.

It should read: “…… and CSC-associated gene expression in CRC cell lines.

Comments on the Quality of English Language

is o.k.

Reviewer 3 Report

Comments and Suggestions for Authors

 MACC1 regulates LGR5 to promote cancer stem cell properties in colorectal cancer

By Müge Erdem, Kyung Hwan Lee, Markus Hardt, Joseph Regan, Dennis Kobelt, Wolfgang Walther, Margarita Mokrizkij,  Christian Regenbrecht and Ulrike Stein.

The authors show that the metastasis inducing MACC1 also serves as transcriptional regulator for the G protein-coupled receptor LGR5 in colon cancer cells. LGR5 induces stem cell traits in CRC cells. The transcriptional induction of LGR5 by MACC1 was tested in human CRC cell lines.

The authors demonstrate that high MACC1 expression correlated with high LGR5, high S100A4, a MACC1 downstream target, high ALDH1 activity and with functional readouts of stemness, such as colony formation and tumor sphere formation. MACC1 KD in CRC cells resulted in decreased LGR5 and NANOG and decreased functional stemness readouts. ChIP assays revealed the binding of MACC1 to the LGR5 promoter in SW620 cells. Further, MACC1 expression correlated with CRC stem cell markers CD44, NANOG and LGR5 in patient-derived tumor organoids (PDO). The authors show a significant correlation between MACC1 and LGR5 gene expression in their own and other publicly available CRC patient cohorts. The authors provide evidence for a new role of MACC1 as transcriptional regulator for LGR5 and demonstrate that MACC1 binds to the promoter region of the LGR5 gene. Thus, they have identified a role for the metastasis associated MACC1 as CRC stem cell marker and therapeutic target in CRC.

The discovery of MACC1 as a transcriptional regulator of the stem cell factor LGR5 reveals yet another mechanism by which MACC1 promotes stem cell-driven CRC metastasis.

The manuscript is well written, the methodology is appropriate to answer the questions and the data are sound.

There are a few minor corrections that should be addressed before publishing:

Figure 3b: significance is not indicated in the graphs

Relative mRNA expression is shown in Figures 1-3. Have the authors detected MACC1 protein for some of these assays to confirm the presence of the MACC1 protein? It would be important to show presence of the MACC1 protein in some of the patient-derived cells to strengthen the finding that MACC1 serves as TF for LGR5 and its stem cell functions.

Fig 4b: What are the MACC1 counts (Y-axis label) from in-vivo xenografts? Are these counts of MACC1 positive cells normalized to the total number of total cells in the xenograft section? How many sections were investigated? It needs to be explained what the counts are normalized to.

Fig.5 legend: forced re-expression of MACC1 in SW620 cells only rescued LGR5, not NANOG levels – this needs to be corrected in the text and the figure legend and it would help to suggest distinct gene regulation (if known) for NANOG and LGR5 in the discussion section.

Fig. 6b: the reduced sphere formation may be caused by reduced proliferation of the cells. Cell proliferation for both cell models with exogenous expression or KO of MACC1 needs to be determined and related to the sphere formation changes.

Fig. 8b: the ChIP graph should be better labelled for the X-axis so readers don’t have to visit the method section to understand which antibodies have been used for IP versus detection – this is not clear from the graph. The antibodies should be listed in the method section.
